# Adapting genetic algorithms for artificial evolution of visual patterns under selection from wild predators

**Emmanuelle S. Briolat** [1]*, **George R. A. Hancock**[1,2], **Jolyon Troscianko**[1]

1 Faculty of Environment, Centre for Ecology and Conservation, Science and Economy, University of Exeter, Penryn, Cornwall, United Kingdom, 2 Faculty of Biological and Environmental Sciences, University of Helsinki, Helsinki, Finland

* esb204@exeter.ac.uk

**Data Availability Statement:** Code for tools used in this paper can be found on GitHub (https://github.com/GeorgeHancock471/CamoWild_Repository_CE.v1.2). The tools used for creating printed targets, including site randomisation, have

## Abstract

Camouflage is a widespread and well-studied anti-predator strategy, yet identifying which patterns provide optimal protection in any given scenario remains challenging. Besides the virtually limitless combinations of colours and patterns available to prey, selection for camouflage strategies will depend on complex interactions between prey appearance, background properties and predator traits, across repeated encounters between co-evolving predators and prey. Experiments in artificial evolution, pairing psychophysics detection tasks with genetic algorithms, offer a promising way to tackle this complexity, but sophisticated genetic algorithms have so far been restricted to screen-based experiments. Here, we present methods to test the evolution of colour patterns on physical prey items, under selection from wild predators in the field. Our techniques expand on a recently-developed open-access pattern generation and genetic algorithm framework, modified to operate alongside artificial predation experiments. In this system, predators freely interact with prey, and the order of attack determines the survival and reproduction of prey patterns into future generations. We demonstrate the feasibility of these methods with a case study, in which free-flying birds feed on artificial prey deployed in semi-natural conditions, against backgrounds differing in three-dimensional complexity. Wild predators reliably participated in this experiment, foraging for 11 to 16 generations of artificial prey and encountering a total of 1,296 evolved prey items. Changes in prey pattern across generations indicated improvements in several metrics of similarity to the background, and greater edge disruption, although effect sizes were relatively small. Computer-based replicates of these trials, with human volunteers, highlighted the importance of starting population parameters for subsequent evolution, a key consideration when applying these methods. Ultimately, these methods provide pathways for integrating complex genetic algorithms into more naturalistic predation trials. Customisable open-access tools should facilitate application of these tools to investigate a wide range of visual pattern types in more ecologically-relevant contexts.

been condensed and modified for greater ease of use in CamoEvo Version 2.0 and will be included in all future versions. Instructions on using these tools are provided in the CamoEvo 2.0 handbook on GitHub (https://github.com/GeorgeHancock471/ CamoEvo-v2.0-2022_Plugins/tree/main). All other relevant data and analysis code are within the manuscript and its Supporting Information files.

**Funding:** GRAH was funded by a Natural Environment Research Council (NERC) GW4+ studentship (NE/S007504/1), JT & ESB by a NERC Independent Research Fellowship awarded to JT (NE/P018084/1). https://www.ukri.org/councils/ nerc/

**Competing interests:** The authors have declared that no competing interests exist.

## Introduction

Camouflage is a common anti-predator defence strategy, with a long history as a case study for evolution [1]. A vast body of work has investigated how visual patterns help conceal prey from predators, identifying a range of strategies from disruptive coloration and countershading to masquerade and iridescence [1,2]. Understanding which patterns produce the most effective protection is an important goal for camouflage research, particularly with respect to practical applications, yet determining which combinations of traits provide optimal camouflage is not straightforward. The typical approach of comparing the detectability of a set of phenotypes, varying in specific traits, has been hugely successful in determining relevant features for crypsis, but is poorly-suited to exploring camouflage optimisation: the vast phenotypic space available for colours and patterns, along with interacting effects of viewing conditions, backgrounds and receiver vision, make it unfeasible to test all possible combinations in this way [3]. Moreover, prey patterns are shaped by long-term interactions between communities of predators and prey, learning across repeated encounters and co-evolving over evolutionary time. In particular, search image formation, whereby predators learn to more efficiently detect common prey types, is thought to lead to apostatic selection, preventing the evolution of a stable optimum and favouring polymorphisms in prey patterns [4,5].

Inspired by the processes of natural selection, artificial evolution of visual patterns using genetic algorithms presents an alternative approach, allowing for exploration of a wide phenotypic space and accounting for the effects of repeated interactions between predators and prey. Originally developed as a general solution to optimisation problems with vast parameter spaces [6], genetic algorithms are an increasingly popular technique in biological research, ideally suited to exploring visual pattern space. So far, genetic algorithms have been used to investigate the evolution of several aspects of visual signalling, primarily camouflage patterns [4,7–12], but also aposematism [13], motion dazzle [14] and even nectar guides in flowers [15]. In brief, genetic algorithms set out both virtual genetic structures underpinning phenotypes and rules governing survival and reproduction into the next generation, based on the responses of an observer; across multiple encounters with the observer in successive generations, optimal solutions to the task should arise. When the observer is a foraging predator, prey items that are easiest to detect are selected against, and so lost from the population, while those that are most difficult to find will survive and breed, ultimately leading to the evolution of better-camouflaged patterns.

Following similar principles, early experiments in virtual ecology tracked populations of different-coloured pastry baits under selection from wild birds, with the proportion of different bait types in each generation determined by the ratio of survivors in the previous round [16,17]. Building on this simple paradigm, genetic algorithms incorporate processes of mutation and sexual reproduction, allowing for more complex variation in prey phenotype, and for novel traits to arise. In addition to the algorithms determining how patterns are retained, lost or modified between generations, the ways in which different pattern traits are encoded is also increasingly sophisticated. While no models claim to replicate the actual genetic architecture of a particular species, advances in our understanding of how animal patterns are genetically determined and physically produced during development [18] have inspired systems better able to produce a range of realistic natural patterns [12].

So far, these more complex genetic algorithms have generally been applied only in computer-based experiments, where they are most easily implemented, but which fail to capture the impacts of naturalistic viewing conditions and relevant receiver vision. Techniques such as projecting the search area on larger screens, using natural background images with occlusion layers, or simulating the visual system of dichromats can increase the realism of the search task

[11], but still fall short of replicating the effects of real predator behaviour. To our knowledge, genetic algorithms have only been combined with selection by non-human predators in one scenario: seminal experiments on frequency-dependent selection in which blue jays (*Cyanocitta cristata*) searched for artificial moths with evolving patterns on screens [4,7]. Besides restricting prey patterns to greyscale targets, these experiments required highly-motivated captive predators—undergoing up to eight months of pre-training then 100 generations' worth of trials [7]—so are difficult to expand to other species and contexts.

Here, we describe a more generalisable process that combines the benefits of field trials involving natural predators with state-of-the-art tools for artificial evolution, expanding on a recently-released genetic algorithm and pattern generation framework, the CamoEvo toolbox [12]. Prey patterns are generated by CamoEvo, with modifications facilitating implementation in the field, and deployed on physical prey items, concealing a food reward. Wild predators then interact with these items in semi-controlled arenas, and the order in which prey are attacked guides the evolution of subsequent generations of patterns. As a proof-of-principle, we present a case study in which we tested for effects of background complexity on the evolution of camouflage patterns, in artificial prey under selection from wild birds. We also replicated the field trials in computer-based experiments with human volunteers; this allowed us to compare the performance of the new field-based methods to well-established screen-based protocols, and explore how field experiments could be rapidly prototyped and parametrised in future applications.

## Materials and methods

### General principles

We developed a methodology for generating and testing artificial prey in the field using genetic algorithms, by substantially adapting an existing screen-based implementation, the CamoEvo toolbox [12], released here as CamoPrint in CamoEvo V2.0 (see supporting code). The properties of the prey and predation scenarios can be customised, and multiple colour and pattern metrics can be used to analyse the outcomes of pattern evolution, providing flexibility to suit a range of research questions (Fig 1). To begin with, an initial population of prey patterns is created by the CamoPrint software, following experimenter inputs for starting parameters, such as prey size, shape, population size and the colour space from which prey colours can be drawn (see [12] and supporting code). This first generation of patterns is applied to real physical objects and deployed where predators of interest can interact with them. Suitable prey items can take virtually any form required, so long as they can conceal a food reward accessible to predators, or otherwise provide a record of predation attempts. We describe a technique for transferring computer-generated patterns onto plasticine or 3D-printed shapes (see case study), but appropriate two- and three-dimensional targets can also be made more simply by printing on paper (Fig 1). The setting, visual backgrounds and predators recruited will depend on the experimental question; these methods could be implemented in a wide range of scenarios, from relatively controlled predation trials with captive animals to more naturalistic experiments with multiple predators and prey scattered across a large area. As predators interact with prey, the timing or order of predation is recorded, whether through direct observation, filming, or regular checks, as in more traditional artificial predation experiments. Each population of prey can also be split into discrete groups, for example if prey are to be presented in a series of semi-controlled arenas (see case study). In that case, prey can be ranked according to the order of attack within each group; the size and number of groupings, and prey arrangement within arenas, can be customised as needed. Predation outcomes are then fed back into the genetic algorithm software, which uses this information to determine which patterns

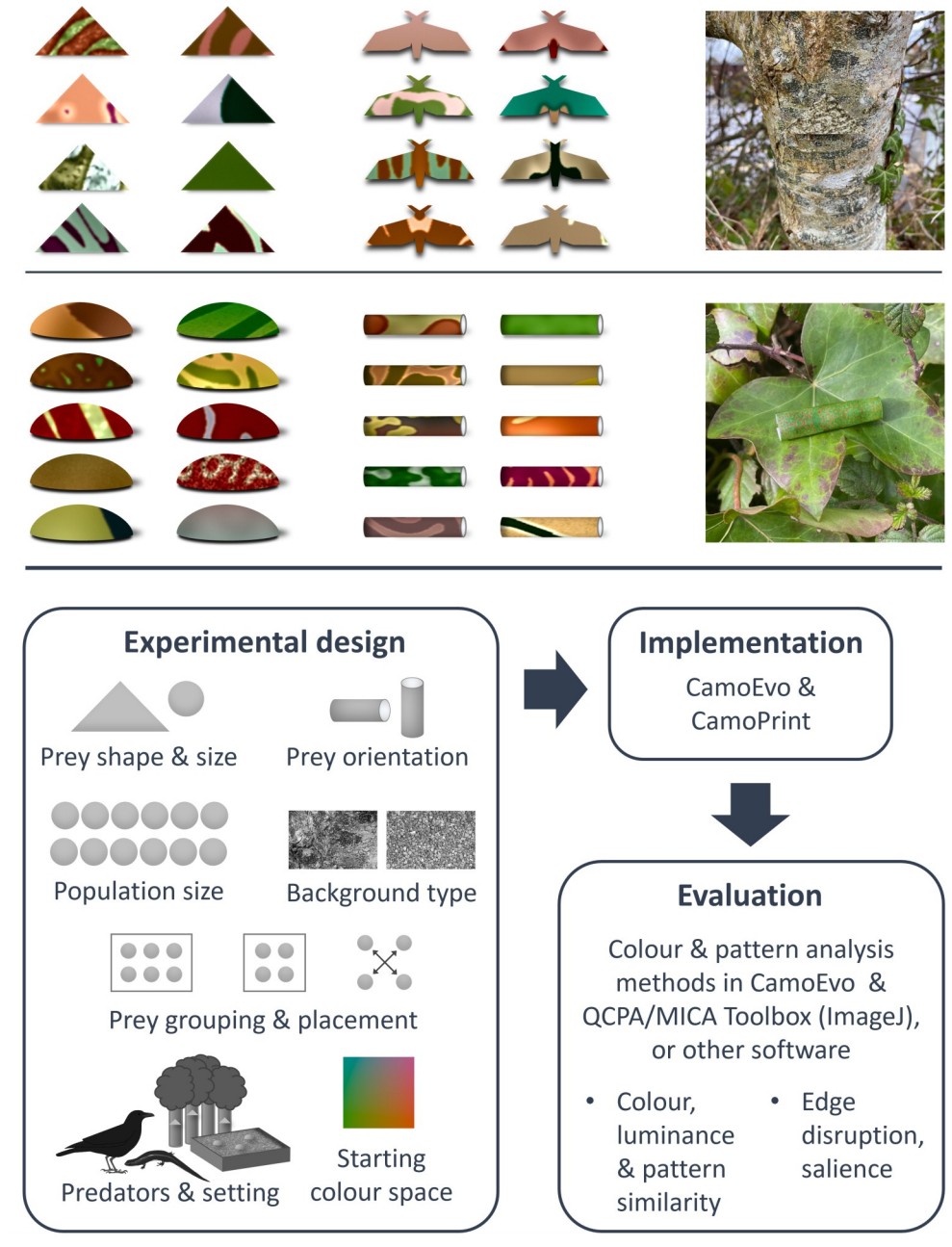

**Fig 1. Examples of possible target types and general methods workflow.** Users can customise a range of experimental parameters, and investigate multiple pattern traits when analysing results. Prey presented here include paper targets, two-dimensional with simple or more specific shapes (top), and three-dimensional rolled paper tubes, as well as 3D-printed shells (middle, as used in case study). Rightmost panels show example prey *in situ*.

survive and reproduce into the next generation. To facilitate the integration of physical prey items into the existing CamoEvo workflow, we implemented several modifications to the pattern generation tools–to create the target shapes and shading used in the case study, assign prey to groups and positions in test arenas, and more easily input the results from predation trials–which can be adapted to suit each user's needs (see Supporting Information (SI) for details, and supporting code). Repeating these processes of pattern generation, prey

preparation, predation trials and data entry allows experimenters to follow selection by predators across multiple generations of prey.

Finally, to explore the outcome of selection, experimenters can select a wide range of colour, luminance and pattern metrics to analyse changes in prey appearance across generations. CamoEvo can analyse the computer-generated prey patterns using a range of tools in the widely-used MICA and QCPA toolboxes in ImageJ [19,20], providing a fully open-access workflow, from experimental design to the assessment of pattern evolution. Alternatively, the physical prey items themselves can be photographed and processed using the same open-access tools, so that image analyses can account for the real 3D properties of prey and backgrounds, and relevant visual systems.

## Case study—prey pattern evolution against backgrounds of varying three-dimensional complexity

As a proof-of-concept, we present an application of these methods, investigating the evolution of camouflage patterns in response to selection from wild birds. We deployed prey in arenas with specific visual backgrounds, to test the effect of differences in three-dimensional background complexity on the strength and direction of pattern evolution. Theoretical models [21] and experiments with both human and avian predators [22–24] suggest even relatively poorly-camouflaged targets are more difficult to detect against complex backgrounds, relaxing selection for increased similarity to the background [24]. In addition, three-dimensional backgrounds introduce variation in lighting conditions, between shadowed areas and those under direct lighting, which creates noise in both luminance and colour [25,26]. As a result, we expected to see improved camouflage in our targets across generations, but with a weaker signal of directional change in the population of prey displayed against backgrounds with greater three-dimensional complexity.

**Field set-up.** Two sets of artificial predation experiments were run in different locations in the grounds of the University of Exeter's Penryn campus (United Kingdom; 50˚ 10' 12" N, 5˚ 7' 22.8" W), from 17th September to 21st October and 10th November to 11th December 2021. Experiments were approved by the University of Exeter's ethics committee (Application ID: eCORN003896 v3.1).

Our set-up was designed to encourage natural foraging behaviour by wild birds, while allowing us to control the backgrounds prey were viewed against, and accurately track predation events. Prey were placed in six wooden trays (53x53cm, 22cm above the ground), set out in areas where birds likely to interact with the set-up were regularly seen. Each tray was lined with a polypropylene ground cover membrane, allowing water to filter through, and a mix of gravel and construction sand, forming the visual background (Fig 2). Many alternative backgrounds could be used here; this gravel mix was selected as a relatively natural and heterogeneous option, that was also stable enough to remain in place in poor weather conditions and could be sculpted to vary the complexity of the scene. To test for an effect of background complexity, each experimental run monitored the survival of two populations of prey, presented in trays in which the gravel was either smoothed flat or given 3D structure, with wavy furrows creating peaks and troughs in various orientations (henceforth, smooth and furrowed treatments; Fig 3).

Each population consisted of 24 prey, divided into six groups of four per tray. Every morning, half of each population was placed out into three trays with smooth or furrowed gravel respectively, so each generation of both populations was fully exposed to predators across two consecutive days (Fig 3). The location of the smooth and furrowed backgrounds among the trays was alternated daily, and the prey positions in each tray were pseudo-randomised using a

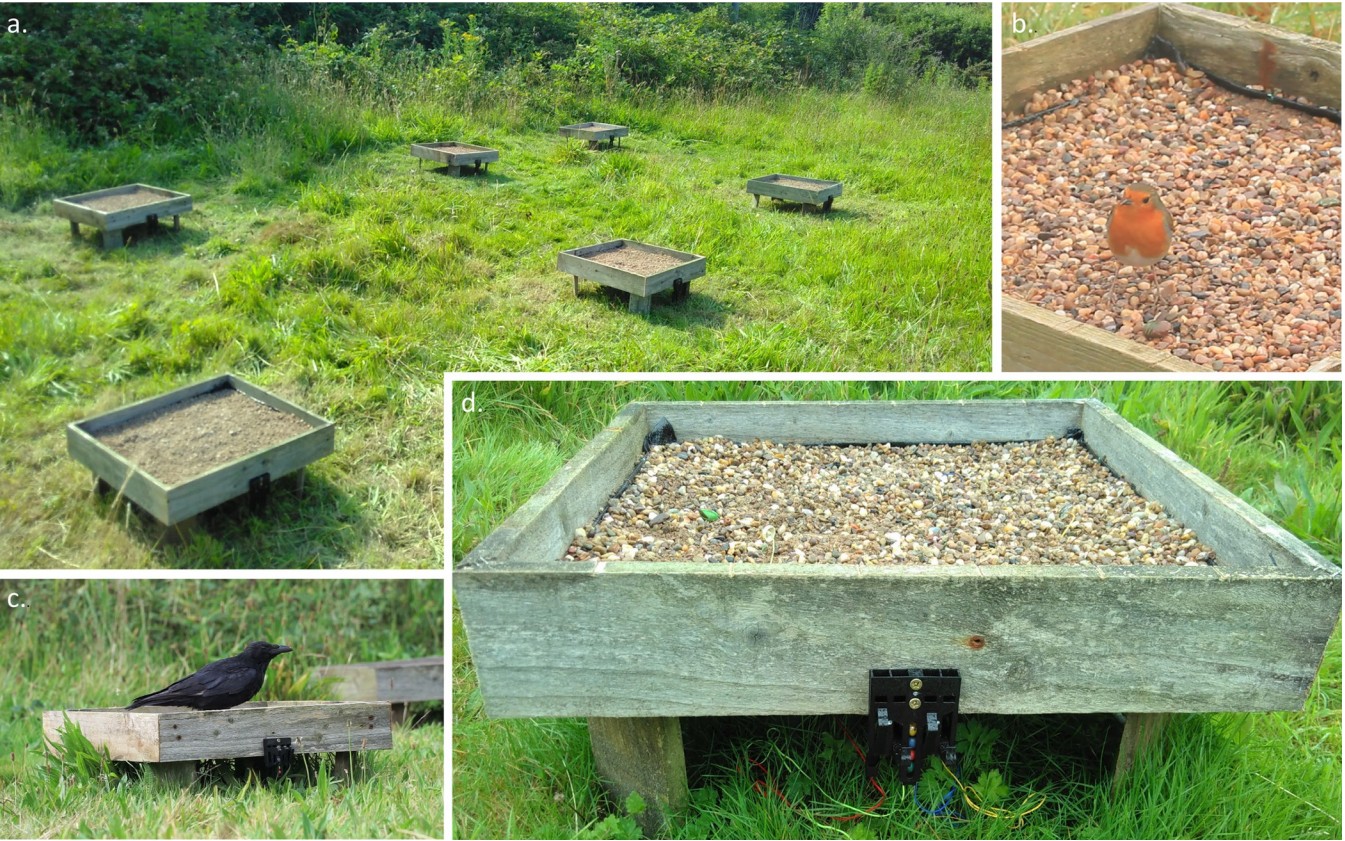

**Fig 2. Experimental set-up for field trials.** Tray array for the first run (a), with the main predators involved, *E. rubecula* (b) and *C. corone* (c), and a side view of a tray and mechanical timing gate (d). Here, the first two items attacked were associated with the green and red marbles (leftmost prey).

custom script in ImageJ [27], designed to reduce spatial clustering. In CamoPrint, custom additions to the CamoEvo toolbox automated the allocation of prey patterns to each tray and labelled them for easier identification (see SI and supporting code). The artificial prey items themselves consisted of 3D-printed oval shells (18x14 mm) which birds had to flip over to access a concealed food reward (dried mealworm and suet). Prey pattern was manipulated by covering the shells in a thin (~1mm) layer of white tack (UHU, Bühl, Germany), then applying custom patterns printed on waterslide transfer paper (Rolurious) with a laser printer (Colour Laser Jet Enterprise M553, HP, USA). Printing on waterslide paper is a practical way of printing on thin transparent film; after briefly soaking in water, the transparent film separates from its paper backing, and adheres strongly to the white tack surface. Printing targets in CIELAB space does not allow for manipulation of the UV component of colour patterns, which could be relevant to tetrachromatic avian vision; however, printing target patterns on paper with low UV reflectance is a widespread technique for artificial predation experiments testing camouflage strategies with avian predators (e.g. [22]). In this instance, both the targets and gravel backgrounds had low UV reflectance, and the printer could produce colours that matched patches of the background substrates even as perceived by tetrachromatic models of avian vision ($\Delta S < 1$, see results, Fig 4 & S1 Data).

Prior to the experimental trials, wild birds were gradually attracted to the trays and trained to search for the artificial prey, starting with a single tray with visible baits, moving on to food in upturned prey shells, then concealed rewards, and finally increasing the number of trays

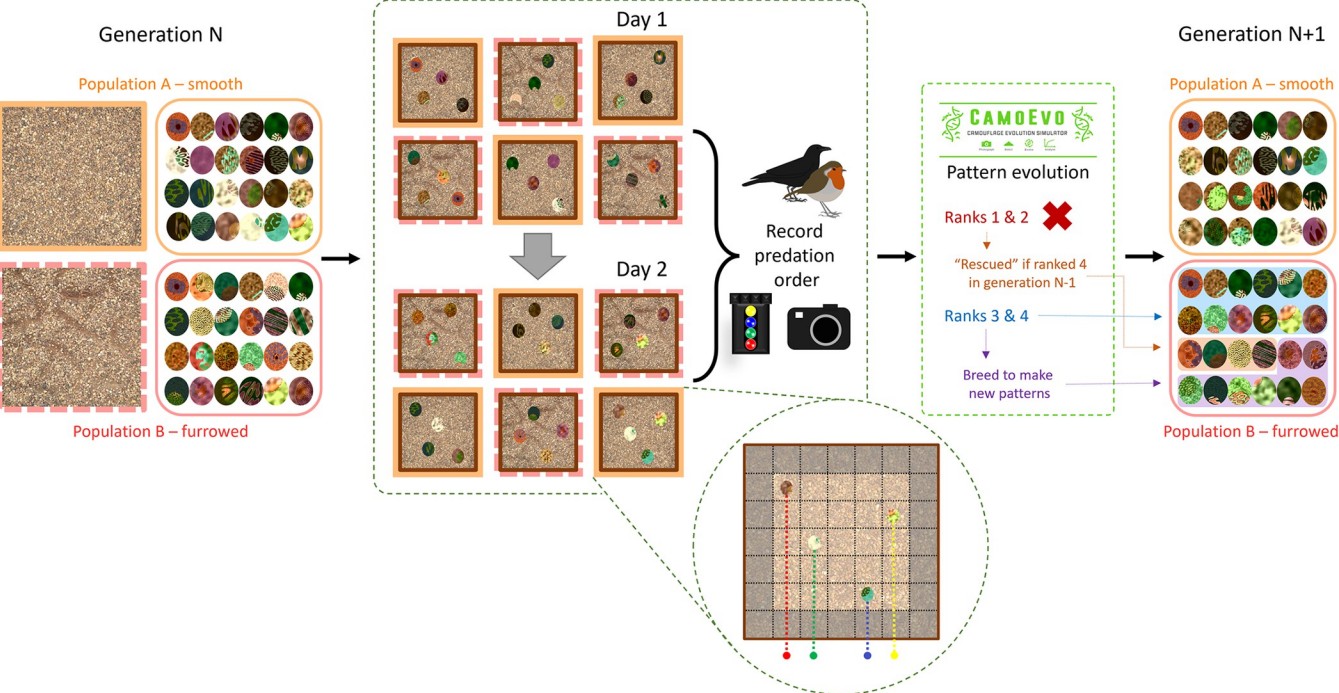

**Fig 3. Schematic of the process for selection and evolution of prey patterns in each generation.** Inset: Diagram representing prey positioning in the trays, with strings connecting to the mechanical timing gate.

and prey. During training, prey patterns consisted of checkerboards with colours sampled from the colour space of the experimental patterns. Trials began once visiting birds had consumed four prey in all six trays for at least three consecutive days. In the first run, the trays were visited by carrion crows (*Corvus corone*); in the second, a robin (*Erithacus rubecula*) was the main predator, with occasional visits from blackbirds (*Turdus merula*) and great tits (*Parus major*) (Fig 2; S1 Fig in S1 File; S1 Video).

During experimental trials, the order in which prey items were attacked in each tray served as our measure of prey detectability. To automatically record this, the prey shells were tethered from the underside to coloured strings running under the trays, connecting them to a mechanical timing gate. The four prey items in each tray were always assigned string colours in the same order–red, green, blue and yellow from left to right–with marbles of matching colours in the timing gate; when an item was attacked, its string was pulled, releasing a marble of the corresponding colour to create a visible record of attack order (Figs 2 and 3). This system was not always sufficiently sensitive to record attacks from smaller birds, so as a fail-safe, each tray was also filmed continuously with a small spy camera (Coulax).

Prey were set out each morning (8 - 9am), and trays were checked every two to three hours; each trial ended once at least three items had been attacked per tray. If no prey were attacked over the course of the day, the trays were covered up overnight, and the same prey presented again the following day. Once each generation had been fully exposed to predation, the order of prey attacks in every tray was entered into the genetic algorithm software, and a new set of prey patterns was produced for the next generation. See Fig 3 for a full schematic of the experimental procedure.

**Pattern generation and algorithm set-up.** Prey patterns were generated using the CamoEvo toolbox's animal pattern generator, with a modified script for the 3D targets used in

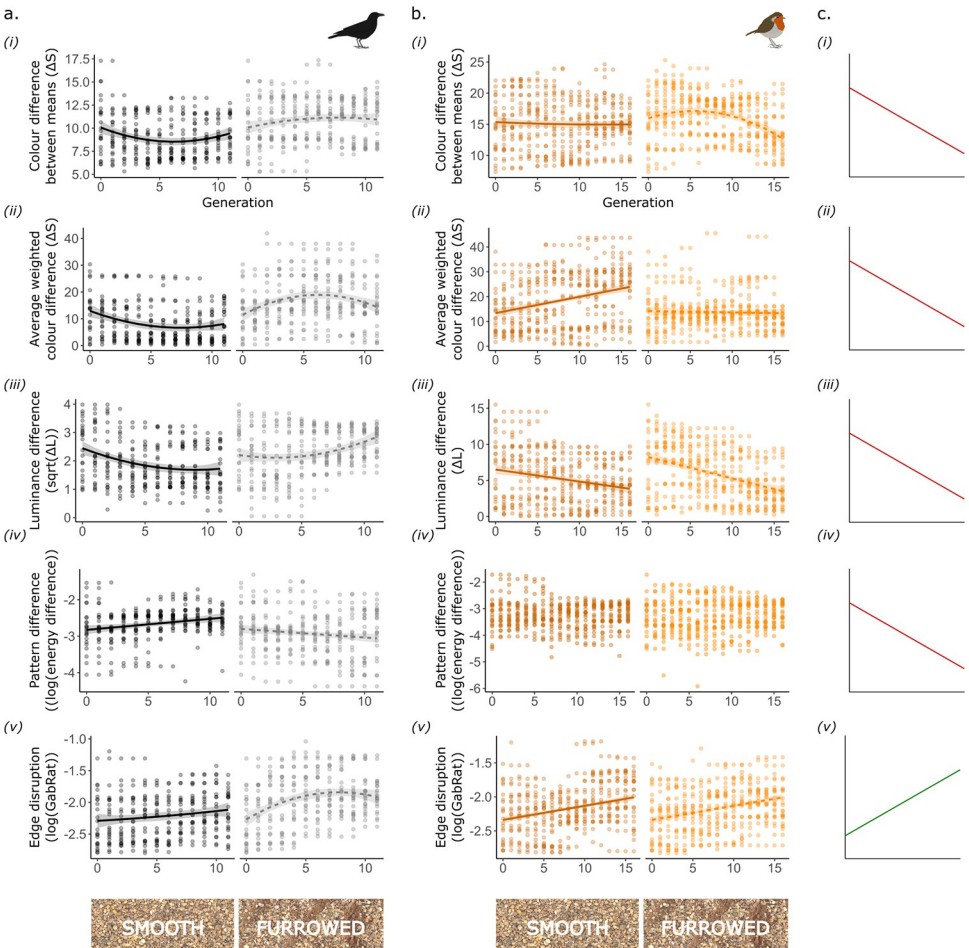

**Fig 4.** Metrics of camouflage efficacy across generations, in targets from the first (a) and second run (b) of field trials, alongside expected trends for improved camouflage (c). All plots display targets shown against smooth backgrounds in the left panel, and furrowed backgrounds on the right. Trend lines and shaded ribbons represent final model predictions, with 95% confidence intervals. Generation was always initially fitted as a second-degree polynomial; non-linear trends are shown if the polynomial term was significant in final models.

this experiment (see SI). Targets were oval, with colourations occupying a CIELAB space with ranges 0–100 in luminance, -50–50 in A (green-red) and -10–70 in B (blue-yellow). Every population began with an identical set of patterns, randomly-generated using these parameters with a uniform distribution of genes (Generation 0; S2 Fig in S1 File). Details of the pattern generation and genetic algorithm framework can be found in [12].

CamoEvo is primarily designed for computer-based experiments, so we implemented several modifications to facilitate its application to a field experiment with physical prey items, where prey survival must be manually reported (described in full in SI). These modifications form the basis of the CamoPrint tools featured in CamoEvo V2.0. Once the trials for each generation were complete, prey were assigned a fitness ranking from 1 to 4 in every tray, based on the order in which they were attacked. The genetic algorithm then selected the last two patterns attacked in each tray (ranks 3 & 4) to survive and reproduce, creating the next generation of patterns. However, to reduce the likelihood of relatively well-camouflaged prey being eliminated by chance, we applied CamoEvo's built-in rescue system [12]: if a prey pattern was ranked 1 or 2 in generation N, but ranked 4 (the highest possible rank) in generation N-1, the

pattern also survived into the next generation (N+1). In the first run, trials stopped after generation 10, when crows stopped visiting the site; in the second, trials ended as planned after 15 generations. To account for the final set of predator decisions, we used CamoPrint to produce an additional generation of patterns (generation 11 or 16 respectively), included in subsequent analyses.

Prey camouflage was assessed with digital photography and image analysis, using well-established methods implemented in the Multispectral Image Calibration and Analysis (MICA) and QCPA plugins for ImageJ [19,20]. The following commonly-used metrics of camouflage efficacy were analysed, accounting for perception by relevant avian visual systems: chromatic ($\Delta S$) and achromatic differences ($\Delta L$) between mean colours in the Receptor Noise-Limited space [20,28,29], pattern difference based on granularity analyses [30], and target edge disruption, measured using the GabRat method [31]. To account for prey pattern, where component colours may better match the background than the mean prey colour, we also calculated a frequency-weighted average measure of colour difference between all colours in each prey pattern and the most similar colours in the background. Statistical analyses were carried out in R version 3.5.2 [32]. See SI for details.

**Screen-based search tasks.** The outcomes of the field trials suggested our methods could be improved by refining the experimental parameters, to adjust task difficulty. To investigate this, we replicated the field trials with matching screen-based search tasks with human volunteers, allowing us to more rapidly test the impacts of different starting conditions; compared to a month-long run of field trials, each replicate of the screen-based task took roughly only half an hour for a participant to complete. These search tasks were also designed using the CamoEvo toolbox, and run using its built-in user interface for screen-based psychophysics experiments. All parameters were selected to mimic the field experiment as closely as possible, with the exception of starting colour space: target patterns occupied either the same colour space as prey in the field (henceforth the full colour space), or a narrower colour space, corresponding to 2 standard deviations around the mean colour of the background images. In the initial generation 0, targets from the full colour space were thus identical to generation 0 targets in the field, while those from the narrow colour space featured the same patterns but with a restricted range of colours (S5 Fig in S1 File). Each participant viewed a single population of targets over 16 generations, shown on photographs of the same smooth or furrowed gravel backgrounds as in the field. Every image contained four targets to find, and participants were tasked with clicking on them as quickly as possible, within a maximum of 15 seconds per slide. Survival of targets into the next generation and pattern evolution were governed by the same rules as in the field, based on the order in which targets were clicked. Changes in the detectability of targets over successive generations were quantified using capture time, the time taken for participants to click on each target, as recorded by the CamoEvo toolbox. Measures of prey phenotype used to assess camouflage efficacy, and statistical analyses, were also selected to match those used for prey in the field trials as closely as possible (see SI for details of the experimental design and analyses). In addition, we used CamoEvo to produce 12 replicate populations, with identical starting conditions to the experimental populations, but in which prey patterns were selected at random to survive and reproduce into the next generations. These provide control trends, against which experimental populations can be compared to demonstrate the effect of selection by human volunteers, as opposed to neutral drift.

A total of 12 volunteers completed these screen-based tasks, between 23rd March and 13th April 2022 (three per combination of background type [smooth or furrowed] and colour space [full and narrow]). Experiments took place in a dark room, with images displayed on a 27" monitor (iiyama, Japan). Participants were recruited among staff and students at the University of Exeter, aged 18–70 with normal or corrected-to-normal vision and not knowingly

colour-blind. The experiments complied with the declaration of Helsinki and received ethical approval from the University of Exeter's CLES Cornwall Ethics committee (application ID: 512348, date approved: 22/03/22). Participants received an information sheet presenting the rationale and rules of the search task, and signed a consent form to approve the use of their anonymised data in this study.

## Results

### Case study—pattern evolution in field trials

Prey populations showed evidence of improved camouflage across generations, though results varied between background types and experimental runs (Fig 4). In the first run, colour and luminance differences between prey patterns and the background decreased, as expected, for the population shown on smooth gravel, but actually increased slightly for the population on furrowed gravel (linear model (lm), generation:background, $F_{2,570} = 7.625$, $F_{2,570} = 17.425$, $p < 0.001$, for colour difference between means and the weighted average colour difference respectively; $F_{2,570} = 18.997$, $p < 0.001$ for luminance; Fig 4A(i-iii)). Conversely, pattern differences decreased across generations in the more complex furrowed treatment, but increased in the smooth (lm, generation:background, $F_{1,572} = 16.282$, $p < 0.001$; Fig 4A(iv)), suggesting that the relative importance of pattern and colour matching varied between backgrounds. In the second run, luminance difference decreased in both populations, but most strongly on the furrowed backgrounds, (lm, generation:background, $F_{1,812} = 6.869$, $p = 0.009$; Fig 4B(iii)). However, there was little improvement in colour-matching: differences between mean target and background colours declined across generations only on the furrowed background (lm, generation:background, $F_{2,810} = 10.761$, $p < 0.001$; Fig 4B(i)) while the weighted average measure found little change in that population, and an increase in colour difference across generations on the smooth background (lm, generation:background, $F_{1,812} = 28.069$, $p < 0.001$; Fig 4B(ii)). Unlike in the first run, there were no significant trends in pattern difference (lm, generation:background, $F_{2,810} = 1.328$, $p = 0.266$; background, $F_{1,812} = 1.524$, $p = 0.217$; generation, $F_{2,813} = 0.975$, $p = 0.378$; Fig 4B(iv)). When improvements in background-matching were observed, effect sizes were small, and targets remained conspicuous, with colour and luminance differences well above accepted detection thresholds of 1 to 3 $\Delta S$ or $\Delta L$. For example, in the smooth background population in the first run, colour differences between means declined from 10.28 $\Delta S$ in generation 0 to 9.32 $\Delta S$ in generation 11 (Fig 4A(i)). Luminance differences reached lower levels, suggesting that luminance matching was more critical for camouflage, but this could be due to lower starting values; for example, in the same population, mean luminance difference started at 7.07 $\Delta L$ in generation 0, reaching 3.47 $\Delta L$ by generation 11 (Fig 4A (iii)).

Most consistently, targets in all populations displayed increased edge disruption across generations (Fig 4A, 4B(v)); this improvement appeared to level off in the population shown on furrowed backgrounds in the first run (lm, generation:background type, $F_{2,570} = 4.754$, $p = 0.009$), but not in the second (lm, generation:background, $F_{2,810} = 1.976$, $p = 0.139$; background, $F_{1,812} = 0.531$, $p = 0.467$; generation, $F_{1,814} = 72.172$, $p < 0.001$).

### Evolution in screen-based replicates

In the screen-based experiments, camouflage metrics reveal clear differences between populations of prey utilising the full and narrow colour spaces. For populations using the narrow colour space, patterns improved in every camouflage metric across generations (Fig 5A). There were significant decreases in luminance, colour and pattern differences between targets and the backgrounds (lme, for luminance: generation, $\chi^2_1 = 114.07$, $p < 0.001$; for colour:

generation, $\chi^2_2 = 191.31$, p < 0.001; for pattern: generation, $\chi^2_1 = 8.466$, p = 0.004), and an increase in edge disruption in target patterns (lme, for GabRat in L: generation, $\chi^2_1 = 8.043$, p = 0.005), all with no significant differences between populations shown against smooth or furrowed backgrounds (see S1 Table in S1 File). For the weighted average colour difference metric, there was a significant effect of the interaction between generation and background type, describing different patterns of overall decrease in colour differences (generation:background type, $\chi^2_2 = 7.434$, p = 0.024; Fig 5A(iii)). These improvements in target camouflage made it harder for participants to find them, as suggested by a significant increase in the time taken to capture the first target on each slide over successive generations for the narrow colour space, regardless of background type (lme, generation: background, $\chi^2_2 = 4.762$, p = 0.093, background: $\chi^2_1 = 3.826$, p = 0.050, generation: $\chi^2_2 = 6.340$, p = 0.012; Fig 5A(vi)).

By contrast, for the full colour space, there was substantial variation in the trajectory of pattern evolution between replicate populations, making it difficult to see consistent evidence of selection for improved camouflage (Fig 5B). There were no significant trends in luminance or pattern difference between the targets and the background across generations and no evidence of increased edge disruption in prey patterns (Fig 5B(i), (iv-v), S2 Table in S1 File). There was however some evidence for improved colour-matching overall: average colour difference between targets and backgrounds decreased slightly across generations (lme, generation, $\chi^2_2 = 7.255$, p = 0.027), and using the weighted average colour difference metric suggests that this improvement was greater against the smooth backgrounds (lme, generation:background, $\chi^2_2 = 6.965$, p = 0.031, Fig 5B(ii-iii), S2 Table in S1 File). Capture time was shaped by the interaction between generation and background type, but did not increase overall (lme, generation: background, $\chi^2_2 = 10.832$, p = 0.004; Fig 5B(vi)).

Results from the control populations, simulating the effects of neutral drift, confirm that consistent selection for improved camouflage was only seen in experimental populations using the narrow colour space. Under those conditions, there were significant differences between the trajectories of experimental populations, selected by volunteers, and control replicates across generations: average colour and luminance differences between target patterns and their backgrounds decreased in the selected populations, while they increased slightly in the controls (lme, generation:scenario, for luminance $\chi^2_1 = 9.443$, p = 0.002, for colour, $\chi^2_1 = 15.557$, p < 0.001; S6 Fig in S1 File). Similar trends for the weighted average colour difference metrics and edge disruption were not significant, suggesting that volunteers in our experiment were selecting primarily for improved colour and luminance match (see S6 Fig in S1 File, S3 Table in S1 File). However, for the full colour space, analysing all replicates together revealed no significant interaction between scenarios (control versus experimental) and generation, for any metrics (see S6 Fig in S1 File, S4 Table in S1 File).

## Discussion

This study presents methods to support investigations of camouflage evolution using genetic algorithms in the field, with prey items exposed to wild predators. Our detailed case study describes how to implement these tools, from modifications of the genetic algorithm software to techniques for making artificial prey and recording predation events, and demonstrates the feasibility of this approach. Free-flying birds reliably interacted with multiple generations of prey, and prey patterns improved in several metrics of camouflage efficacy, suggesting the potential for adaptive evolution.

However, this case study also highlights important factors to consider when designing this type of evolutionary experiment. Trends for improved camouflage in the field were relatively weak, and inconsistent among runs with different predators and background types. We tested

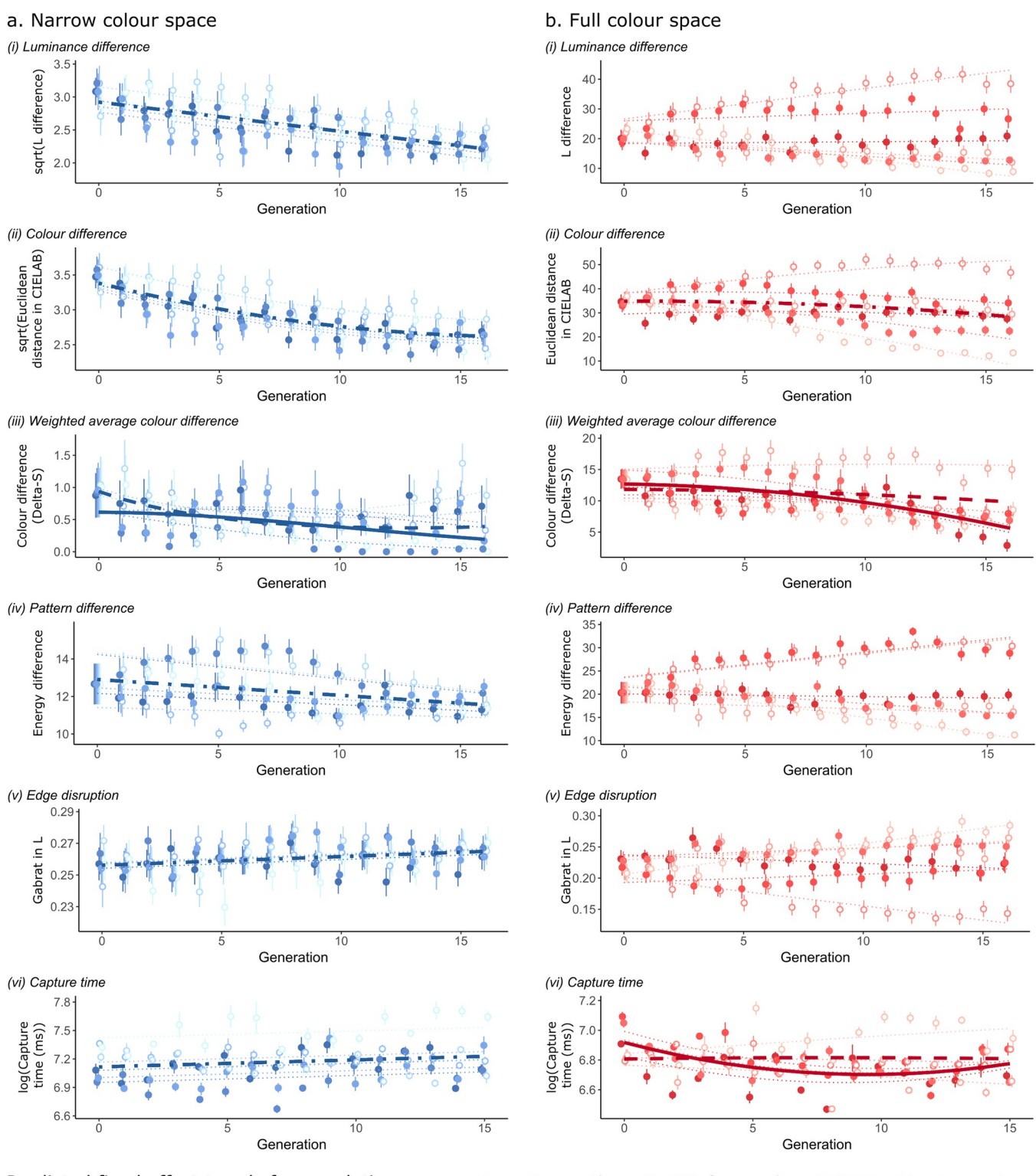

**Fig 5.** Changes in camouflage metrics in the screen-based experiments, for prey populations using the narrow (a) and full (b) colour spaces. Plots show the mean and standard error of each metric per generation, along with predictions from the best statistical models. Shades of each colour represent different populations, with filled dots for those shown on smooth backgrounds, and open dots for furrowed backgrounds. Dotted lines represent population-level predictions, and thicker lines overall trends for fixed effects only.

whether these findings were linked to the starting colour space available for prey patterns, using screen-based search tasks with human volunteers, which closely replicated the field trials and were much quicker to run. These experiments revealed substantial variation in outcomes produced by individual participants, when target patterns utilised the same colour space as in the field: while camouflage improved across generations in some populations of prey, others became more conspicuous, such that few overall trends for improved camouflage were detected, and populations were not significantly different from control replicates, simulating neutral drift. These results were unexpected, as similar psychophysics experiments in artificial evolution, with starting parameters very close to the full colour space used here, typically find much clearer trends towards improved camouflage [12]. However, the number of prey and the way they are presented strongly affect learning in this type of experiment. Previous screen-based experiments with background-matching, disruptive or distractive camouflaged prey revealed that learning varied among human participants depending on whether targets were shown sequentially or simultaneously, and the number of prey shown together. For example, detection rates for high-contrast disruptive prey decreased faster than for background-matching prey when they were presented simultaneously, while there was no difference in learning over trials with sequential presentations of single prey [33]. Predator strategies, such as the formation of search images for specific prey types versus searching for more general features, will thus vary depending on the search scenario [33]. This will affect which prey will be detected first, shaping the evolutionary trajectories of prey patterns, so that outcomes are likely to be different in our multiple-prey paradigm compared to more traditional scenarios where prey are presented singly and sequentially. Here, the presence of multiple prey per image increases the influence of random factors, such as the position of prey relative to one another. Under these circumstances, participant learning outstripped target evolution, highlighting the unexplored power of predator learning in holding back the evolution of camouflage from a conspicuous evolutionary start point, a scenario that may be relevant to aposematic species, for example. Predator cognition, encompassing learning, neophobia, forgetting, recognition errors and adaptive decision-making, is a key consideration in the evolution and maintenance of aposematism as a signalling strategy [34,35]. Theoretical models suggest that elements of both predator psychology and prey ecology, such as prey density and aggregation, can contribute to the initial evolution of aposematism [36], and our set-up presents opportunities to investigate these effects, by varying the profitability and arrangement of prey in both the field and online systems. Predator behaviour and cognition are similarly important for the evolution of camouflage yet have received less attention in this context [37], so further experiments with real predators could also shed more light on the conditions required for crypsis to evolve, given different predator communities. In our experiments, the screen-based trials suggest the trajectory of evolution is essentially random when prey utilise the full colour space, so any specific results from the field case study should be interpreted with caution.

By contrast, the screen-based experiments showed that populations of prey utilising a narrower colour space, based on colours found in the backgrounds, were not only more cryptic from the start, but also consistently evolved to be better camouflaged over successive generations, and clearly behaved differently from control replicates. We therefore predict that field trials would similarly yield greater improvement in camouflage if the prey patterns started with a more restricted colour space, based on the colours of the background. In addition, very few metrics of camouflage efficacy revealed differences in prey pattern evolution between populations displayed against the smooth and furrowed backgrounds, even in the screen-based experiments with the narrow colour space. In the field runs, trends did vary between background types, but there was little consistency across metrics, or between repeats, and thus no clear sense that selection was stronger for populations on the simpler smooth background.

This suggests that the variation in three-dimensional complexity between these backgrounds was also insufficient to select for different evolutionary trajectories in this scenario. As such, these experiments represent a proof-of-principle for virtual evolution in the wild, but future applications should carefully consider starting parameters, to better calibrate the difficulty of the task.

Researchers attempting similar work would benefit from testing their starting parameters in pilot experiments with human volunteers as "predators", to establish conditions in which robust selection can be expected to take place before running more time-consuming experiments in the field. To that end, the methods described here present a significant advantage, as CamoEvo's flexibility allows users to set up screen-based experiments that closely mirror their field setups, and rapidly collect data with human volunteers. Equally, the limitations of our field experiments highlight the difficulty of predicting the conditions under which prey patterns can be expected to evolve, which is in of itself an interesting area to explore. Using these field methods to investigate the scenarios that favour the evolution of camouflage in response to real predators would be extremely valuable, to better understand the role of predation as a selective force. In particular, the impact of the starting colour space available to prey has received little attention, yet is likely to be a significant constraint for the subsequent evolution of prey patterns. Beyond prey appearance, initial conditions are an important determinant of evolutionary trajectories more generally. For example, certain traits can promote or restrict the potential for adaptive radiation (eg. when competition between small individuals for a shared resource limits the emergence of larger individuals that could specialise on alternative resources [38]) while more evolutionarily labile traits can facilitate the spread of invasive species, or boost the resilience of native species they encounter [39]. So-called "historical difference experiments", testing how populations that previously evolved under different conditions perform in a new situation, provide a means to explore the role of starting conditions and historical contingencies in evolution [40]. Our set-up provides opportunities to run this kind of experiment, potentially by combining screen and field experiments for efficiency, and approach these broad evolutionary questions through the study of pattern evolution.

The system we present here is designed to encourage future research on prey appearance, by providing open-access tools for every step of the process, from pattern generation and the genetic algorithm framework [12], to implementation in the field, and image analysis software for quantifying changes in prey properties [19,20]. Within CamoEvo and CamoPrint, parameters such as target shape, colour space and genetic algorithm rules are fully customisable, while the field set-up can be modified to meet individual needs, for different predators and backgrounds. Like all predation experiments relying on printed targets [eg. 22,41], matching colourations can be complicated, particularly if the observers can perceive a broader range of colours than humans (e.g. are sensitive to UV wavelengths). Moreover, the range of colours that can be reliably printed is somewhat limited; prey in these experiments could not, for example, evolve colours that were outside the printer's colour gamut, or iridescence through structural coloration. As with all genetic algorithms, the parameters chosen for pattern generation and the operation of the genetic algorithm itself will define which traits can be subject to selection, so these should be carefully selected when designing experiments, and considered when interpreting results. However, acknowledging these limitations, the printing techniques shown here are relatively flexible, as they can be creatively applied to a range of materials, allowing for potential manipulation of UV reflectance if required. In addition, while CamoEvo's built-in tools provide comparisons between target and background colours for human vision only, physical targets and backgrounds can be photographed post-hoc, and image analyses applied to estimate visual contrasts accounting for the full range of predator vision. These methods should therefore facilitate the application of virtual evolution to investigating a wide

range of visual pattern types and functions. In particular, identifying optimal camouflage patterns remains a key question, with many practical applications. Recent studies tackling this problem have gained useful insights by defining optimal patterns based on the statistical properties of the background [41], or using deep learning to extrapolate from human responses to a limited set of phenotypes [3]. Yet genetic algorithms present an especially powerful tool for studying camouflage optimisation, particularly when paired with machine learning techniques [10,11]. Crucially, they account for the effects of repeated interactions between predators and prey, though implementation in screen-based experiments restricts their interpretation. Associating these tools with predation experiments in the field provides opportunities to investigate optimal camouflage while simultaneously considering a wider range of influential factors, from background properties and viewing conditions to predator vision, learning and prey population dynamics. Beyond camouflage, field-based evolutionary approaches supported by these tools could also provide valuable new insights into other forms of anti-predator colouration, such as aposematism or distant-dependent signalling.

## Supporting information

**S1 Data.**
(ZIP)

**S1 File. SI1. Additional methodological information for case study field trials. S1 Fig. Evidence of predation from a crow (*C. corone*) in run 1 (a), and a robin (*E. rubecula*), great tit (*P. major*) and blackbird (*T. merula*) in run 2 (b-d).** Still frames extracted from footage captured by camera traps. **SI 2. Additional information for screen-based search tasks. S4 Fig. Target image modification for screen-based replicate experiments.** Unmodified target (left), and target as viewed by volunteers (right), after shadow and gloss layers have been applied to the target and a drop shadow has been applied to the background image. **S5 Fig. Schematic of the screen-based search tasks, showing the targets in generation 0 and 15, for populations using the full and narrow colour spaces.** Populations from both colour spaces were shown against either smooth or furrowed backgrounds, with three replicates per combination. Each replicate was completed by a single participant. For generation 15, each column of targets represents the six most successful targets (slowest to be found) for each replicate population. **S6 Fig. Changes in camouflage metrics in the screen-based experimental populations and control replicates, using the narrow (a) and full (b) colour spaces.** Plots show the mean and standard error of each metric per generation, coloured by population, along with predictions from the best statistical models. Coloured circles represent experimental populations, greyscale triangles control runs. Dotted lines represent population-level predictions; thicker lines show overall trends for fixed effects only. **S1 Table. Model simplification tables for full models testing changes in metrics of camouflage efficacy in the screen-based experiment, for the narrow colour space.** Models include population-level random slopes for generation, except for luminance and colour difference (a-b), edge disruption (e) and capture time (f). Models with a polynomial term for generation are fully simplified, then the usefulness of the 2$^{nd}$ order polynomial term is tested with an ANOVA, comparing final models with and without the polynomial terms. Significant factors are highlighted in italics. **S2 Table. Model simplification tables for full models testing changes in metrics of camouflage efficacy in the screen-based experiment, for the full colour space.** Models include population-level random slopes for generation. Models with a polynomial term for generation are fully simplified, then the usefulness of the 2$^{nd}$ order polynomial term is tested with an ANOVA, comparing final models with and without the polynomial terms. Significant factors are highlighted in italics. **S3 Table. Model simplification tables for full**

**models testing changes in metrics of camouflage efficacy between control and experimental populations of the screen experiment, for the narrow colour space.** Models include population-level random slopes for generation, except for models of edge disruption. Models with a polynomial term for generation are fully simplified, then the usefulness of the $2^{nd}$ order polynomial term is tested with an ANOVA, comparing final models with and without the polynomial terms. Significant factors are highlighted in italics. **S4 Table. Model simplification tables for full models testing changes in metrics of camouflage efficacy between control and experimental populations of the screen experiment, for the full colour space.** Models include population-level random slopes for generation, except for models of edge disruption. Models with a polynomial term for generation are fully simplified, then the usefulness of the $2^{nd}$ order polynomial term is tested with an ANOVA, comparing final models with and without the polynomial terms. Significant factors are highlighted in italics.
(DOC)

**S1 Video. Example predations in field trials.**
(MPG)

## Acknowledgments

We thank Juho Jolkkonen & Andrew Szopa-Comley for fieldwork assistance, all screen experiment volunteers, Will Allen for helpful comments on an earlier version, and two anonymous reviewers.

## Author Contributions

**Conceptualization:** George R. A. Hancock, Jolyon Troscianko.

**Formal analysis:** Emmanuelle S. Briolat.

**Investigation:** Emmanuelle S. Briolat, George R. A. Hancock.

**Methodology:** Emmanuelle S. Briolat, George R. A. Hancock, Jolyon Troscianko.

**Supervision:** Jolyon Troscianko.

**Visualization:** Emmanuelle S. Briolat, George R. A. Hancock.

**Writing – original draft:** Emmanuelle S. Briolat.

**Writing – review & editing:** Emmanuelle S. Briolat, George R. A. Hancock, Jolyon Troscianko.

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
