## [Decision Letter · Decision Letter 0]

6 Mar 2024

PONE-D-23-37346Adapting genetic algorithms for artificial evolution of visual patterns under selection from wild predatorsPLOS ONE

Dear Dr. Briolat,

Thank you for submitting your manuscript to PLOS ONE. After careful consideration, we feel that it has merit but does not fully meet PLOS ONE’s publication criteria as it currently stands. Therefore, we invite you to submit a revised version of the manuscript that addresses the points raised during the review process.

We look forward to receiving your revised manuscript.

Kind regards,

Matthew Shawkey

Academic Editor

PLOS ONE

Journal Requirements:

"GRAH was funded by a Natural Environment Research Council (NERC) GW4+ studentship (NE/S007504/1), JT & ESB by a NERC Independent Research Fellowship awarded to JT (NE/P018084/1). 

" ext-link-type="uri" xlink:type="simple">https://www.ukri.org/councils/nerc/"

"We thank Juho Jolkkonen Andrew Szopa-Comley for fieldwork assistance, all screen experiment volunteers, and Will Allen and an anonymous reviewer for helpful comments on an earlier version. GRAH was funded by a NERC GW4+ studentship (NE/S007504/1), JT ESB by a NERC Independent Research Fellowship awarded to JT (NE/P018084/1)."

"GRAH was funded by a Natural Environment Research Council (NERC) GW4+ studentship (NE/S007504/1), JT ESB by a NERC Independent Research Fellowship awarded to JT (NE/P018084/1). 

" ext-link-type="uri" xlink:type="simple">https://www.ukri.org/councils/nerc/"

**Additional Editor Comments:**

I apologise for the length of time this review process has taken. It was difficult to find willing reviewers and then one reviewer ultimately let us down after a long period of waiting. We have obtained one review and I have read the paper myself. I agree with the reviewer that this is an interesting and well-done paper, but also that the effects of uv coloration need to be taken into account and discussed in the manuscript. I look forward to receiving your revision.

Reviewers' comments:

Reviewer's Responses to Questions

**Comments to the Author**

1. Is the manuscript technically sound, and do the data support the conclusions?

Reviewer #1: Yes

2. Has the statistical analysis been performed appropriately and rigorously? 

Reviewer #1: Yes

3. Have the authors made all data underlying the findings in their manuscript fully available?

Reviewer #1: Yes

4. Is the manuscript presented in an intelligible fashion and written in standard English?

Reviewer #1: Yes

5. Review Comments to the Author

Reviewer #1: In this manuscript the authors present an extension to the well-established field, laboratory, and computational methods that are widely used in studies of colour evolution and function. By developing a novel experimental and computational protocol they link field predation studies, screen-based detection experiments, and genetic algorithms. This protocol is tested using physical targets and wild birds and then compared to a similar experiment using digital targets and human participants. The main focus of the manuscript is to describe and test the methodological approach, in this regard the two protocols (field and screen-based) do find similar results when the same starting parameters (i.e., target colour space) are used (although starting conditions are shown to affect the results in the screen-based experiments).

I think the authors have done an excellent job at explaining their study in a clear, engaging, and concise manner in both the main text and supplementary material. I only really have one main question about the methodology/interpretation and a couple of minor comments on the text.

Firstly, the colour space used throughout the field experiment is CIELAB L*a*b* which does not extend into the ultraviolet. Did the background or any visible part of the targets (e.g., the white tack if visible) reflect UV? If so, could the inability of the printer/colour space to replicate UV (or near UV for VS birds) reflectance have rendered all targets highly detectable? For example, is it possible that if UV contrast is high for all targets selection on L*a*b* colours may have been reduced? I think this could be discussed in the Methods and/or SI1.

Similarly, how may a mismatch between the printed colour space and the full range of visible colours affect the success of these evolutionary field studies? And are there any mitigations/cautions that you recommend be considered when designing these treatments/experiments (i.e., only use where UV will have a negligible effect)? I think limitations of the approach should be examined in the Discussion.

Minor comments:

Lines 101-103: Does the replication with humans also allow for assessment/benchmarking of the novel approach relative to established protocols? If you agree, I think you could add this as an additional benefit of your approach.

Line 213: I am not familiar with waterslide transfer paper, but I assume this is not just sticking waterproof paper to the white tack - can you explain that this is a transfer more explicitly please (here or in the SI)?

Lines 440-442: Do these similar psychophysics experiments use the full or a restricted colour space?

Lines 445-448: It is not clear to me how your data relate to predator learning. Here, are you referring to observer learning in your dataset or describing the results presented in citation 33 (Troscianko et al. 2013. PLoS ONE)?

6. PLOS authors have the option to publish the peer review history of their article (what does this mean?). If published, this will include your full peer review and any attached files.

Reviewer #1: No

---

## [Author Response · Author response to Decision Letter 0]

17 Apr 2024

We have addressed all editorial and reviewer comments in our cover letter for the resubmitted manuscript, and the "response to reviewers3 document attached with the submission. The cover letter contains a revised funding statement, following guidance from the decision letter. I have also pasted our responses to the reviewer's comments below:

Editor comments: We have obtained one review and I have read the paper myself. I agree with the reviewer that this is an interesting and well-done paper, but also that the effects of uv coloration need to be taken into account and discussed in the manuscript. I look forward to receiving your revision.

Thank you for reading our manuscript, and for your overall positive response. We have addressed the issue of UV vision in our avian predators in response to the reviewer’s comments on this subject, with additional clarification in the methods and discussion sections (see response to reviewer comments below for details).

Reviewer #1: In this manuscript the authors present an extension to the well-established field, laboratory, and computational methods that are widely used in studies of colour evolution and function. By developing a novel experimental and computational protocol they link field predation studies, screen-based detection experiments, and genetic algorithms. This protocol is tested using physical targets and wild birds and then compared to a similar experiment using digital targets and human participants. The main focus of the manuscript is to describe and test the methodological approach, in this regard the two protocols (field and screen-based) do find similar results when the same starting parameters (i.e., target colour space) are used (although starting conditions are shown to affect the results in the screen-based experiments).

I think the authors have done an excellent job at explaining their study in a clear, engaging, and concise manner in both the main text and supplementary material. I only really have one main question about the methodology/interpretation and a couple of minor comments on the text.

We thank you for the generally positive review of our manuscript, and for your comments suggesting further improvements.

Firstly, the colour space used throughout the field experiment is CIELAB L*a*b* which does not extend into the ultraviolet. Did the background or any visible part of the targets (e.g., the white tack if visible) reflect UV? If so, could the inability of the printer/colour space to replicate UV (or near UV for VS birds) reflectance have rendered all targets highly detectable? For example, is it possible that if UV contrast is high for all targets selection on L*a*b* colours may have been reduced? I think this could be discussed in the Methods and/or SI1.

The patterns of the printed targets used in the field experiment were generated in CIELAB space by the CamoEvo software, but we then photographed the target patterns and backgrounds with bandpass filters allowing us to measure reflectance in the visible and UV range, and analysed colour contrasts using tetrachromatic models of bird vision (see Supplementary Information). This means that the contrast values between targets and backgrounds we provide in the results section account for any UV reflectance of the targets and backgrounds, and the full range of avian vision. Results from the new average colour difference metric, based on segmenting the target patterns and backgrounds into different colour patches, show close matches between at least some printer colours and background patches (ΔS1), demonstrating that the printer can produce colours that closely match the background, even accounting for UV (see figure 4a,b(ii) and supporting data). In this case, both the gravel backgrounds and targets have low UV reflectance. We have amended the text to explain this in the methods section (l. 218-225).

Similarly, how may a mismatch between the printed colour space and the full range of visible colours affect the success of these evolutionary field studies? And are there any mitigations/cautions that you recommend be considered when designing these treatments/experiments (i.e., only use where UV will have a negligible effect)? I think limitations of the approach should be examined in the Discussion.

The explanation in the methods section should already address some of these concerns, highlighting why the absence of control over the UV component of target patterns was not a substantial limitation in our case study. However, this is a really valid point: in general artificial predation experiments are somewhat limited by the ability to produce only a restricted range of colours, in this case based on the printer’s colour gamut. We have added a short section in the discussion to address the potential limitations of printing patterns, as well as to highlight how some of these difficulties can be overcome, or at least accounted for – by using different materials that do allow higher UV reflectance if this is required, and at least by photographing the actual prey items to analyse their coloration using models that do account for the full range of predator vision (see l. 533-548). 

Minor comments:

Lines 101-103: Does the replication with humans also allow for assessment/benchmarking of the novel approach relative to established protocols? If you agree, I think you could add this as an additional benefit of your approach.

Thank you for the suggestion – we have added in a note to that effect (l.102-103). 

Line 213: I am not familiar with waterslide transfer paper, but I assume this is not just sticking waterproof paper to the white tack - can you explain that this is a transfer more explicitly please (here or in the SI)?

Waterslide paper is effectively a clear transfer film, bonded to an opaque white backing sheet so that it can easily be used in a standard printer. When the paper is briefly soaked in water, the clear film separates from the backing sheet, and can be placed on the surface of the target item (this can be a layer of white tack, as in our case study, or directly onto a thermoplastic or 3D printed shell). When damp, the clear film adheres well to the new surface, and then becomes very difficult to remove, so it lasts well even in field conditions. We have added a sentence in the methods to explain the process more explicitly (l. 215-218).

Lines 440-442: Do these similar psychophysics experiments use the full or a restricted colour space?

The colour space used in the paper cited here (Hancock Troscianko, 2022) corresponds to CamoEvo’s default settings (CIELAB space with a luminance range of 0, 100, A range of −60, 60, and B range of −10, 70) and is very similar to the full colour space used in the field trials in this present manuscript (L: 0 – 100, A: -50 – 50, B: -10 – 70, see Methods). We have added a note to make that similarity explicit (l. 452-453)

Lines 445-448: It is not clear to me how your data relate to predator learning. Here, are you referring to observer learning in your dataset or describing the results presented in citation 33 (Troscianko et al. 2013. PLoS ONE)?

In this sentence, we are indeed describing the findings in reference 33 (Troscianko et al. 2013); we have now added more description of their results to make this clearer (l. 455-463). Since the same birds are returning day after day to the trays in our field experiments, we expect learning to be an important component of how these birds respond as the experiments progress. In Troscianko et al. (2013), a series of screen-based experiments with human “predators” clicking on targets employing various camouflage strategies (distractive markings, disruptive camouflage, background-matching), trends in detection times across trials revealed differences in how participants improved in detecting the different target types, based on the design of the experiments, i.e. whether targets were shown sequentially or simultaneously on the same screen, and if the latter, how many targets were shown at once. This difference in target presentation is therefore likely to be an important factor in determining the strategies employed by predators searching for the targets; this will affect which targets are found first, hence shaping the evolution of target patterns in our experiment, and likely explaining some of the difference in outcomes between our setup (where multiple prey are shown at once) and comparable experiments in Hancock Troscianko (2022), where targets are shown one at a time. We have now amended the text to more explicitly explain this (l. 455-466).

---

## [Editor Report · Decision Letter 1]

30 Apr 2024

Adapting genetic algorithms for artificial evolution of visual patterns under selection from wild predators

PONE-D-23-37346R1

Dear Dr. Briolat,

We’re pleased to inform you that your manuscript has been judged scientifically suitable for publication and will be formally accepted for publication once it meets all outstanding technical requirements.

Kind regards,

Matthew Shawkey

Academic Editor

PLOS ONE

Additional Editor Comments (optional):

Congratulations on your interesting work!
---

## [Editor Report · Acceptance letter]

3 May 2024

PONE-D-23-37346R1 

PLOS ONE

Dear Dr. Briolat, 

I'm pleased to inform you that your manuscript has been deemed suitable for publication in PLOS ONE. Congratulations! Your manuscript is now being handed over to our production team.

Kind regards, 

on behalf of

Dr. Matthew Shawkey 

Academic Editor

PLOS ONE